# Study on Demethoxycurcumin as a Promising Approach to Reverse Methicillin-Resistance of *Staphylococcus aureus*

**DOI:** 10.3390/ijms22073778

**Published:** 2021-04-06

**Authors:** Qian-Qian Li, Ok-Hwa Kang, Dong-Yeul Kwon

**Affiliations:** Department of Oriental Pharmacy, College of Pharmacy and Wonkwang Oriental Medicines Research Institute, Wonkwang University, Iksan 54538, Korea; aliceqql@163.com

**Keywords:** methicillin-resistant *Staphylococcus aureus*, MRSA, demethoxycurcumin, anti-bacterial activity, synergistic effect, PBP2a, β-lactam antibiotic

## Abstract

Methicillin-resistant *Staphylococcus aureus* (MRSA) has always been a threatening pathogen. Research on phytochemical components that can replace antibiotics with limited efficacy may be an innovative method to solve intractable MRSA infections. The present study was devoted to investigate the antibacterial activity of the natural compound demethoxycurcumin (DMC) against MRSA and explore its possible mechanism for eliminating MRSA. The minimum inhibitory concentrations (MICs) of DMC against MRSA strains was determined by the broth microdilution method, and the results showed that the MIC of DMC was 62.5 μg/mL. The synergistic effects of DMC and antibiotics were investigated by the checkerboard method and the time–kill assay. The ATP synthase inhibitors were employed to block the metabolic ability of bacteria to explore their synergistic effect on the antibacterial ability of DMC. In addition, western blot analysis and qRT-PCR were performed to detect the proteins and genes related to drug resistance and *S. aureus* exotoxins. As results, DMC hindered the translation of penicillin-binding protein 2a (PBP2a) and staphylococcal enterotoxin and reduced the transcription of related genes. This study provides experimental evidences that DMC has the potential to be a candidate substance for the treatment of MRSA infections.

## 1. Introduction

*Staphylococcus aureus* (*S. aureus*) is a virulent opportunistic pathogen that may colonize various mucosal sites of the body, causing manifold infections, including skin and soft-tissue infections, bloodstream infections, toxin-mediated syndromes, and life-threatening diseases [1]. As one of the “ESKAPE” organisms (*Enterococcus faecium*, *Staphylococcus aureus*, *Klebsiella pneumoniae*, *Acinetobacter baumannii*, *Pseudomonas aeruginosa*, and *Enterobacter* species), *S. aureus* is a threat worldwide because it can cause a variety of serious nosocomial infections [2]. In addition, mutations occur in genes that regulate the metabolic activity of *S. aureus* small-colony variants, resulting in the development and persistence of chronic, recurrent, and drug-resistant infections in the host [3]. Methicillin-sensitive *S. aureus* (MSSA) and methicillin-resistant *S. aureus* (MRSA) are the two main types of infections caused by *S. aureus* [4]. Researchers have been committed to studying antibiotics to combat the invasion of *S. aureus*, however, the emergence of MRSA limits the usage of antibiotics, making the treatment of *S. aureus* infections more complicated [5,6]. MRSA is a common Gram-positive bacterium, causing infections in both the hospital and the community, and can be isolated from hospitalized patients and outpatient settings [7]. Since it was first identified as a hospital-acquired pathogen in 1961, MRSA has become the major cause of death in the intensive care unit (ICU) [8]. In the 1990s, community-acquired MRSA (CA-MRSA) strains appeared in young, healthy individuals who lacked the classical risk factors that have previously been in contact with healthcare institutions [9]. The spread of CA-MRSA has basically spread to every region of the world, and it has become the main type of MRSA infection [3]. The resistance of MRSA to β-lactam antibiotics is inducible, which is the result of the expression of two resistant enzymes, β-lactamase, and penicillin-binding protein (PBP) 2a transpeptidase [10].

The β-lactam antibiotics have long been widely used as the cornerstone for the treatment of infections caused by *S. aureus* [11]. The β-lactam antibiotics, including penicillin, cephalosporin, carbapenem, and monobactam, are the most commonly used antibiotics, which account for more than half of the antibiotics prescribed in clinical settings [12,13]. The β-lactam antibiotics are believed to function by interfering with the bacterial cell wall biosynthesis [14]. Because the cell wall peptidoglycan layers of Gram-positive bacteria is thicker than that of Gram-negative bacteria [15], the β-lactam antibiotics generally have higher inhibitory activity against Gram-positive bacteria. Many Gram-negative bacteria produce β-lactamase to hydrolyze the β-lactam bond of antibiotics and render them ineffective, which is the internal mechanism of Gram-negative bacteria against β-lactam antibiotics [16].

Since self-medication with antibiotics is cheap and timesaving, it is highly prevalent among individuals in low- and middle-income countries, but the frequent and improper use of these medicines has facilitated the emergence of methicillin-resistant [17]. The excessive use of β-lactam antibiotics in clinical treatment and animal husbandry has led to the increasing prevalence of antibiotic-resistant *S. aureus*, which is a worrying global clinical problem [18]. Penicillin-binding proteins (PBPs) are essential membrane-bound proteins for the biosynthesis of bacterial cell wall. The β-lactam antibiotics are known to acylate the transpeptidase domain of PBPs in *S. aureus* [19]. The irreversible binding of β-lactam antibiotics to the active site of PBPs makes them chemically inert and prevents peptidoglycan cross-linking, thereby inhibiting bacterial cell wall biosynthesis, and eventually causing cell lysis and death [20]. However, many *S. aureus* have developed resistance to β-lactam antibiotics due to the acquisition of the *mecA* and *mecC* genes [21,22]. The gene *mecA* is harbored on the mobile genetic element so-called staphylococcal chromosomal cassette *mec* (SCC*mec*) and encodes the penicillin binding protein 2a (PBP2a), an alternative transpeptidase with low affinity for virtually all β-lactam antibiotics [23,24]. *mecC* is known as the divergent *mecA* homologue, and *mecC*-encoded PBP2a (PBP2c), which shares only 63% amino acid identity with *mecA*-encoded PBP2a, shows a greater affinity for oxacillin than for cefoxitin [25]. When the native PBPs are inhibited by binding β-lactam antibiotics, the function of PBP2a can replace the activity of PBPs to continue catalyzing the biosynthesis of the cell wall peptidoglycan [26,27]. Thus, the function of PBP2a allows the normal growth and viability of *S. aureus* even in the presence of high concentrations of β-lactam antibiotics that inhibit natural PBPs [28]. In addition, more than 90% of *S. aureus* also produce β-lactamase encoded by the *blaZ* gene, which cleaves the β-lactam ring and causes hydrolysis and inactivation of the β-lactam antibiotic [29].

Conventional medicine plays a critical role in prevention and treatment of various diseases and conditions for centuries [30]. Therefore, the exploration of antimicrobials extracted from natural herbs as an alternative substance for antibiotics is highly anticipated. Turmeric, the rhizomatous plant belonging to *Curcuma longa* Linn (*Zingiberaceae*), has long been used as a natural medicine to treat various diseases [31]. A large repertoire of evaluations have been conducted on the medicinal value of turmeric, and studies have reported that turmeric extract possesses manifold pharmacological activities, such as anti-inflammatory, antioxidant, anti-tumor, and antimicrobial activities [32,33]. Turmeric contains curcumin (CM), demethoxycurcumin (DMC), and bisdemethoxycurcumin (BDMC) as major curcuminoids [34].

Curcumin is a well-known curcuminoid that has obtained more attention and research, but due to its low aqueous solubility, most of CM is unabsorbed after oral administration. The present study focused on DMC (Figure 1), which is an analog of CM, chemically, without a methoxy group directly linked to the benzene ring [35]. Accordingly, this more stable structure makes DMC exhibit better stable activity and aqueous solubility than CM at physiological pH. It has been reported that DMC has a variety of pharmacological activities, including anti-inflammatory, antihypertensive, neuroprotective, antimicrobial, antifungal, antimalarial, and vasodilatory properties [36]. However, the study of DMC in the treatment of MRSA infection is not comprehensive. Thus, we conducted a series of in vitro study to investigate whether DMC has the therapeutic properties of β-lactamase inhibitor or inhibitor of PBP2a in the treatment of MRSA infections.

The present study focused on exploring the potential of the natural curcuminoid DMC as a novel antibacterial agent, aimed to investigate the sensitivity of MRSA to DMC in vitro, elucidate the possible pathogenic mechanism of DMC inhibiting the growth of MRSA, and evaluate the synergistic or alternative potential of DMC for β-lactam antibiotics.

## 2. Results

### 2.1. The Minimum Inhibitory Concentration (MIC) of DMC and β-Lactam Antibiotics

We initially used the minimum inhibitory concentration (MTT) assay to detect the MIC of β-lactam antibiotic oxacillin (OXA), ampicillin (AMP), gentamicin (GEN), and the curcuminoid DMC. The MIC values of antibiotics and DMC against eight *S. aureus* strains are presented in Table 1. According to the results of the MIC values, all three antibiotics showed a significant inhibitory effect on MSSA ATCC 25923 (ATCC, staphylococcal strains from the American Type Culture Collection). However, the concentration of OXA required for complete inhibition of seven MRSA strains ranged from 0.9 to 1000 μg/mL, the MIC value of AMP against MRSA ranged from 1.9 to 125 μg/mL, and the MIC value of GEN against MRSA was 7.8 to 2000 μg/mL. The high MIC values of β-lactam antibiotics confirmed the resistance of MRSA. We found that DMC was able to inhibit the growth of MRSA and displayed a MIC of 62.5 μg/mL for each *S. aureus* strain. The results showed that DMC has the ability to prevent MRSA from growth and suggested the potential of DMC to be further studied as an antibacterial agent.

### 2.2. Synergistic Effects of DMC and Antibiotics Based on Fractional Inhibitory Concentration Index (FICI)

This study explored the synergistic antibacterial effect between plant extracts and antibiotics by combining curcuminoid DMC with the existing β-lactam antibiotics OXA, AMP, and GEN. As shown in Table 2, DMC was able to reduce the MIC of GEN against seven MRSA strains by 4 to 16 fold. Compared with GEN alone, higher antibacterial activity with the lower MIC was observed when GEN was co-treated with DMC, which indicated that the presence of DMC enhanced the effectiveness of GEN. These results demonstrated that combining DMC with GEN at sub-MIC concentrations obviously increased the susceptibility of MRSA to GEN. However, there is no synergy between DMC and OXA. Similarly, the sensitivity of MRSA to AMP is not enhanced by the addition of DMC. The results regarding OXA and AMP were not shown in the current context.

### 2.3. Time–Kill Assay

Based on FICI, Figure 2 depicted the synergism of DMC and GEN on standard MRSA strains (ATCC 33591) and clinically isolated strains (DPS-2) as the logarithm of the number of viable bacteria colonies at different time intervals. The results indicated that the treatment of DMC or GEN alone did not prevent bacterial growth significantly. After 24 h of incubation, the number of viable bacteria was the same as that of the control group without drug treatment, indicating that DMC and GEN almost lost their efficacy. However, the co-treatment of DMC and GEN at sub-MIC concentrations showed obvious bactericidal activity on both standard strains and clinical strains. It is worth mentioning that the standard strains showed stronger sensitivity to the combined use of DMC and GEN, so that there were no surviving bacteria after 16 h of culture. In brief, compared with other groups, the combination group showed the opposite growth trend, which indicates that the combined treatment at sub-MIC concentrations triggered a synergistic effect. The results of the time–kill assay further confirmed the medicinal value of DMC as an antibacterial agent, and suggested that DMC can be used as an auxiliary drug to treat MRSA infections.

### 2.4. Antibacterial Activity with Membrane Permeability or ATP Synthase Inhibitor

In order to fully understand the anti-MRSA ability of DMC, this study observed the antibacterial ability of DMC under conditions of increased cell membrane permeability or decreased ATP levels. According to our experimental results, in the presence of membrane-permeabilizing agents (TX-100 or Tris), DMC did not show a stronger inhibitory effect on the growth of MRSA (results not shown). According to Figure 3, in the presence of 15.625 μg/mL (1/4MIC) DMC, 25 μg/mL DCCD or 125 μg/mL NaN_3_ alone, MRSA maintained its viability. However, when treated DCCD or NaN_3_, the absorbance value at an optical density 600nm (OD_600nm_) decreased to 31.6% and 38.9%, respectively, indicating that the sensitivity of MRSA to DMC is increased upon treatment with ATP synthase inhibitor. These results indicated that the reduction of metabolic capacity can elevate the anti-MRSA ability of DMC.

### 2.5. The Inhibitory Effect of DMC on the Expression of PBP2a Protein and Related Gene in MRSA

We hypothesize that the reversal mechanism of DMC on MRSA resistance is related to the variation of drug-resistant proteins and related genes. Given that PBP2a is a critical determinant of MRSA resistance mechanism, the present study analyzed the ability of DMC to reverse MRSA resistance by detecting the protein expression of PBP2a. According to Figure 4, compared with the untreated control group, DMC substantially inhibited the translation of PBP2a protein at a sub-inhibitory concentration, and antibacterial effect of DMC was obviously stronger than GEN. Since the translation of PBP2a is the result of triggering the transcription of *mecA*, the current research also analyzed the mechanism of DMC against MRSA in terms of detecting the *mec* operon gene encoding PBP2a. Figure 5a illustrated that the inhibitory effect of DMC on *mec* gene complex at sub-inhibitory concentration was significantly higher than that of the control group without drug treatment and the GEN treatment group, and the higher the concentration of DMC, the greater its usefulness in attenuating bacterial viability. In addition, our research also reinforced the notion that DMC has the potential to develop into β-lactamase inhibitors by detecting the transcription of the *bla* gene complex. As shown in Figure 5b, DMC apparently hindered the expression of *bla* operon gene encoding β-lactamase in a dose-dependent manner. Resistance develops due to the appearance of PBP2a and the production of β-lactamase, while our results suggested that DMC has an augmented bactericidal activity against MRSA due to its ability to disrupt the synthesis of PBP2a and the β-lactamase, thereby reversing the resistance of MRSA. In brief, DMC can be used as PBP2a inhibitor and β-lactamase inhibitor to reasonably explain the mechanism of reversing MRSA resistance.

### 2.6. The Inhibitory Effect of DMC on the Expression of SEA and Related Gene in MRSA

As a representative virulence factor produced by MRSA, staphylococcal enterotoxin A (SEA) was detected to analyze whether the inhibitory mechanism of DMC on MRSA is related to *S. aureus* exotoxins. The secretion of SEA by ATCC 33591 exposed to sub-inhibitory concentrations of DMC was determined by western blotting analysis. Figure 6a depicted that DMC reduced the expression of SEA in a dose-dependent manner compared with the untreated group and GEN treated group. This study also performed qRT-PCR analysis on the sea gene encoding SEA, and the results were consistent with western blot analysis. The sea gene was significantly inhibited dose-dependently after the addition of graded sub-inhibitory concentrations of DMC (Figure 6b). The results indicated that the mechanism of DMC inhibiting invasive MRSA infection may be attributed to its blocking of *S. aureus* exotoxins synthesis.

## 3. Discussion

*S. aureus* is one of the most common pathogens causing severe suppurative skin and soft tissue infections, abscesses, pneumonia, endocarditis and bacteremia [37]. Based on the sensitivity to antibiotics, MRSA is defined as *S. aureus* with a minimum inhibitory concentration of oxacillin greater than or equal to 4 μg/mL, while MSSA is sensitive to β-lactam antibiotics [38]. The use of antibiotics is the last resort against drug resistant pathogens. Nevertheless, the more excessive use of antibiotics inevitably leads to the emergence of resistant strains against the last resort agents, which is described as the “antibiotic resistance spiral” [37]. Due to the development of bacterial resistance, β-lactam antibiotics are no longer the first-line antimicrobial agent for the treatment of *S. aureus* infections. Practically, MRSA has become resistant to all available β-lactam antibiotics [39]. In view of the emergence and widespread dissemination of MRSA in the community, there is an urgent need to study a novel and potent therapeutic strategy of this serious public safety disease. Combining plant extracts with antibiotics is an empirical strategy to overcome the resistance of bacteria [40]. In response to the difficult MRSA infection, the present study was designed to study the effect of DMC, a major phytochemical component originated from turmeric, on reversing the resistance of *S. aureus* to methicillin. Our drug susceptibility experiments on MRSA have shown that DMC has a certain effect on inhibiting the growth of bacteria. In addition, the synergistic antibacterial effect between DMC and antibiotic GEN is a valuable discovery, suggesting the medicinal potential of DMC to assist antibiotics in the treatment of MRSA infections. This study also explored the conditions for enhancing the antibacterial effect of DMC. NaN_3_ and DCCD are used as inhibitors of F_0_F_1_-ATP synthase to impede ATP-binding cassette transporters (ABC transporters) by blocking the proton translocation in F_0_ domain of F_0_F_1_-ATP synthase [41,42]. The results showed that the inhibition of DMC on the growth of MRSA could be facilitated by reducing the metabolic level of bacteria with appropriate concentrations of ATP synthase inhibitors (DCCD and NaN_3_), which is conducive to further exploring the optimal antibacterial conditions for DMC.

There are two main mechanisms for MRSA to develop resistance: one is that even if PBPs are inhibited by binding to β-lactam antibiotics, the bacteria can still synthesize cell wall and maintain normal vitality due to the presence of the alternate PBP2a (encoded by *mecA*) that is poorly bind to β-lactam antibiotics [43]. The other is that bacteria produce β-lactamase (encoded by *blaZ*) to make the β-lactam antibiotics ineffective by hydrolyzing the β-lactam ring [44]. PBPs are enzymes that bind to, and are inhibited by, β-lactam antibiotics, which can synthesize the cell wall peptidoglycan on the outer surface of the cytoplasmic membrane [45]. MRSA can produce a 76KDa PBP2a, which is not available in MSSA. The function of PBP2a is equivalent to all the main functions of natural PBPs. The difference is that the ability of PBP2a to form the acyl-enzyme complexes with β-lactam antibiotics is reduced. Therefore, in the presence of such antibiotics, PBP2a can still complete the synthesis of bacterial cell walls, allowing bacteria to survive and develop drug resistance. According to the determination of PBP2a in this study, it can be found that DMC had a stronger inhibitory effect on the protein expression of PBP2a than GEN, implying the potential of DMC to be further developed as a PBP2a inhibitor.

The acquisition of the *mecA* gene encoding PBP2a is a major determinant of β-lactam resistance in MRSA strains. The *mecA* gene resides on a mobile genetic element designated staphylococcal cassette chromosome *mec* (SCC*mec*) [46]. The modular structure of SCC*mec* is composed of two essential elements: the *mec* gene complex, comprising the β-lactam resistance determinant *mecA* and its regulators (*mecI* and *mecR1*), and the *ccr* gene complex, containing cassette chromosome recombinase (*ccr*) genes responsible for cassette mobilization [47]. The transcription of *mecA* is regulated by the DNA-binding repressor MecI and the transmembrane sensor/transducer MecR1. When β-lactam antibiotics bind to the extracellular sensor domain of MecR1, MecR1 is activated [48]. Thus, the intracellular transducer domain dissociates the MecI from *mecA* promoter region binding site and removes its repression of *mecA* [49]. Subsequently, *mecA* begins to transcript and produces a large amount of PBP2a, which makes bacteria resistant. The qRT-PCR results showed that GEN had no inhibitory effect on MRSA resistance genes, which explained the internal mechanism of antibiotics losing antibacterial effects. However, DMC substantially inhibited the gene expression of *mecA* and *mecR1* at sub-inhibitory concentrations, indicating that the inhibition of PBP2a expression by DMC is related to its interference with the transcription of the *mec* operon elements. According to our experimental results, DMC has shown the ability to inhibit the expression of PBP2a, one of the resistance mechanisms of MRSA, in terms of gene transcription and protein translation, revealing the intrinsic pathway and principle of the phyto-compound DMC in the treatment of MRSA infections.

The β-lactamase is encoded by *blaZ* gene, and more than 90% of Staphylococcus isolates possess the *blaZ* gene and its regulatory sequences (*blaI* and *blaR1*) [50]. The expression of *blaZ* gene, which encodes β-lactamase, is regulated by the sensor/transducer protein BlaR1. Similar in function to MecR1, BlaR1 is a transmembrane sensor and signal transducing protein [51]. The extracellular sensor domain of BlaR1 can detect the presence of antibiotics in the environment and transduce the signal to the cytoplasmic side of the membrane, degrading the gene repressor BlaI [52]. The loss of BlaI results in *blaZ* transcription and subsequent induction of β-lactam resistance. In fact, the transcription of *mecA* and *blaZ* is co-repressed by the two regulators *mecI* and *blaI*, which share 60% sequence homology and have similar functions [43,51,53]. The qRT-PCR results of the *bla* operon element indicated that DMC hampered the transcription of *blaZ* and *blaR1* at sub-inhibitory concentrations, and the 1/2MIC DMC showed a salient inhibitory effect, which once again proved the antibacterial mechanism that DMC prevented the survival of MRSA by inhibiting MRSA resistance genes. As mentioned earlier, β-lactamase and PBP2a are the two main resistance mechanisms of MRSA. According to our in vitro experimental results, DMC had a significant inhibitory effect on both of the resistance determinants, which laid a foundation for us to further explore the anti-MRSA mechanism of DMC.

The staphylococcal enterotoxins (SEs) produced by *S. aureus* are considered to be bacterial superantigens that can induce cell proliferation, toxic shock syndrome, and cytokine storm [54]. In addition, SEs have the potential to cause human staphylococcal gastroenteritis and food poisoning due to its resistance to gastrointestinal proteases and emetic activity [55]. The staphylococcal enterotoxin A (SEA) is the most common cause of staphylococcus-related foodborne poisoning [56]. According to the measurement and analysis of SEA transcription and translation expression, DMC had significant inhibitory effects on the SEA gene and protein expression at sub-inhibitory concentrations dose-dependently, which could be inferred that DMC reduced the vitality of MRSA by suppressing the production of exotoxin induced by MRSA infection. We believe that DMC has more than these anti-MRSA mechanisms explored in this study, and further experiments are needed to fully understand the activity of DMC to reverse methicillin resistance. We will continue to conduct research on DMC to strengthen its feasibility as a substitute for antibiotics to solve MRSA infection.

## 4. Materials and Methods

### 4.1. Bacterial Strains and Growth Conditions

ATCC 33591 standard MRSA strain and ATCC 25923 MSSA strain were purchased from the American Type Culture Collection (Manassas, VA, USA). CCARM 3090, 3091, 3095, and 3102 strains were provided by the Culture Collection of Antimicrobial Resistant Microbes (National Research Resource Bank, Seoul, Korea). Moreover, the remaining two clinical MRSA isolates (DPS-1 and 2) were collected from two different patients at the Department of Plastic Surgery, Wonkwang University Hospital (Iksan, Korea). All test bacteria were incubated on Mueller Hinton agar (MHA) or Brain Heart Infusion agar (BHIA) and suspended in Mueller Hinton broth (MHB) or Brain Heart Infusion broth (BHIB), grown at 37 °C. All strains were stored in 30% glycerol and frozen at −80 °C.

### 4.2. Reagents and Instruments

Mueller Hinton agar, Brain Heart Infusion agar, Mueller Hinton broth, Brain Heart Infusion broth, and skim milk were obtained from Becton, Dickinson and Company (Sparks, MD, USA). Glycerol was purchased from Junsei Chemical Co., Ltd. (Tokyo, Japan). Demethoxycurcumin (DMC), thiazolyl blue tetrazolium bromide (MTT), triton X-100 (TX-100), tris (hydroxymethyl) aminomethane ACS reagent (Tris), N,N’-dicyclohexylcarbodiimide (DCCD), sodium azide (NaN_3_), oxacillin (OXA), ampicillin sodium salt (AMP) and gentamicin sulfate (GEN) were purchased from Sigma-Aldrich Co. (St. Louis, MO, USA). SMART™ bacterial protein extraction solution was purchased from Intron Biotechnology Inc. (Seongnam, Korea). Mouse anti-PBP2a antibody was purchased from DiNonA Inc. (Seoul, Korea). Polyclonal rabbit anti-staphylococcus enterotoxin A antibody ab 15897 was purchased from Abcam (UK). Mouse anti-GAPDH antibody was purchased from Santa Cruz (Dallas, TX, USA). E.Z.N.A.^®^ bacterial RNA kit was purchased from OMEGA Bio-Tek (Norcross GA, USA).

### 4.3. Determination of Minimal Inhibitory Concentration

The present study tested the susceptibility of MSSA and MRSA strains by measuring the minimal inhibitory concentration (MIC). The MIC values of DMC and the β-lactam antibiotics OXA, AMP, and GEN against MRSA and MSSA were determined via broth microdilution assay. A series of 2-fold dilutions with an initial concentration of 1000 μg/mL of DMC were prepared in MHB and BHIB using a 96-well microplate. As for the three antibiotics, the initial concentration was 2000 μg/mL respectively. The inocula were adjusted to 1.5 × 10^6^ colony-forming unit (CFU)/mL. After incubating for 24 h at 37 °C, added MTT reagent and continued incubating for 30 min. The yellowish MTT solution was reduced to a dark blue formazan product by the mitochondrial dehydrogenases of living cells. The color depth is highly positively correlated with the number of live bacteria, which can visually display the minimum inhibitory concentrations.

### 4.4. Determination of the In Vitro Effects of Combinations of DMC and Antibiotics

One approach to treat MRSA infections is to combine existing β-lactam antibiotics with DMC. Thus, the present study investigated the synergism of DMC with OXA, AMP, and GEN using checkerboard dilution test. Mix the serial 2-fold dilutions of DMC with different gradient concentrations of OXA, AMP, and GEN into a 96-well plate. The inocula were adjusted to 1.5 × 10^6^ CFU/mL and incubated at 37 °C for 24 h. The synergistic interaction between the drugs was quantified by fractional inhibitory concentration index (FICI), which was calculated using the following formula: FICI = (A)/MICA + (B)/MICB, where (A) and (B) represent the MIC of drug A and B in combination, respectively; MICA and MICB refer to MIC values of drug A and B alone, respectively. The FICI is interpreted as the following: <0.5, synergy; 0.5–0.75, partial synergy; 0.75–1, additive effect; 1–4, no effect; >4, antagonism.

### 4.5. Time–Kill Assay

The kill kinetics of antimicrobial activity provides a more accurate description of antimicrobial agents, in contrast to the MIC assay [57]. In order to determine the synergistic effect from the survival curve of bacteria, the standard strain ATCC 33591 and clinical strain DPS-2 were used to conduct experiments, and three treatment modes (DMC alone, GEN alone and GEN combined with DMC) were designed to compare the antibacterial effects with the control (no drug). The inocula were diluted to 1.5 × 10^6^ CFU/mL and incubated at 37 °C. At five different time intervals (0, 4, 8, 16 and 24 h), the surviving bacteria were properly diluted and inoculated on the plate. After incubation at 37 °C for 24 h, counted the colonies on the plate. Then the number of viable bacteria was calculated according to the dilution ratio, and the growth curve was drawn.

### 4.6. Determination of the In Vitro Effects of DMC on Membrane-Permeabilizing Agents and ATP Synthase Inhibitors

In order to explore whether the antibacterial effect of DMC can be enhanced by increasing membrane permeability or decreasing level of Adenosine triphosphate (ATP), the sensitivity of MRSA to DMC was determined in the presence of detergents and ATPase inhibitors. In this study, TX-100 and Tris were used as bacterial membrane-binding agents to regulate the permeability of bacterial cell membrane [58], and DCCD and NaN_3_ were used as inhibitors of F_0_F_1_-ATP synthase to block the activity of multidrug resistance efflux pumps [59]. According to the checkerboard dilution test, the serial 2-fold dilutions of DMC dilutions and different gradient concentrations of TX-100, Tris, DCCD, and NaN_3_ were mixed into a 96-well plate. Adjusted the bacteria to 1.5 × 10^6^ CFU/mL and incubated at 37 °C for 24 h. The results were read at OD_600nm_.

### 4.7. Western Blot Analysis

The western blot assay was performed according to the manufacturer’s instructions to analyze protein translation levels. The MRSA (ATCC 33591) suspensions (OD_600nm_ value of 0.7) were treated with graded sub-inhibitory concentrations of DMC and GEN. After shaking culture, the bacterial cells were harvested and suspended in bacterial protein extraction solution. Soluble protein was obtained by centrifuging the bacterial lysates at 13,000 rpm for 10 min. Sodium dodecyl sulfate-polyacrylamide gel electrophoresis (SDS-PAGE) was performed to separate denatured protein (20 μL). The electrophoresed gels were transferred onto polyvinylidene difluoride (PVDF) blotting membranes. The membranes were blocked in 5% skim milk for 2 h, and then incubated with first antibody (anti-PBP2a and anti-SEB) overnight at 4 °C. After incubation with secondary antibody for 1 h, the immunoreactive proteins were detected by TOPview^TM^ ECL Femto Western Substrate (Enzynomics, Korea). ImageQuant LAS-4000 mini chemical luminescent imager (GE Healthcare Life, Korea) was used to visualize the immunoreactive protein bands of membranes.

### 4.8. Quantitative Reverse Transcription Polymerase Chain Reaction (qRT-PCR)

The qRT-PCR was performed according to the manufacturer’s instructions to analyze mRNA transcription levels. The ATCC 33591 strain was grown at an OD_600nm_ of 0.7 in MHB and treated with sub-inhibitory concentrations of DMC and GEN. After shaking culture, centrifuged at 13,000 rpm for 10 min to pellet bacterial cells. Total RNA was isolated using the E.Z.N.A.^®^ bacterial RNA kit according to the manufacturer’s protocol. The mRNA purity were evaluated by measuring the absorbance ratio at 260 nm and 280 nm using a NanoDrop spectrophotometer (Bio-Tek, Winooski, VT, USA). The QuantiTect reverse transcription kit (Qiagen, Seoul, Korea) was used to synthesize the complementary DNA (cDNA) according to the manufacturer’s protocol. Mixed Power SYBR^®^ Green PCR master mix (Applied Biosystems, Foster City, CA, USA), primers, and cDNA. The qRT-PCR was run to synthesize DNA template using the StepOnePlus real-time PCR system (Applied Biosystems, France). The primer sequences are presented in Table 3.

### 4.9. Statistical Analysis

All data were expressed as the mean ± standard deviation (SD) of triplicate tests. The statistical differences between groups were determined with one-way analysis of variance (ANOVA) using statistical software (IBM SPSS version 24). Multiple mean comparisons between treatments were performed using Duncan’s multiple range test (DMRT). Different letters denote statistical differences between different groups (*p* < 0.05), while the same letters mean no significant differences (*p* > 0.05).

## 5. Conclusions

The present study endeavored to investigate the ability of DMC to suppress the pathogenic attack of MRSA. In order to achieve this, we designed a series of drug susceptibility tests to detect the inhibitory effect of DMC on different MRSA strains and the synergistic effect with β-lactam antibiotics, and detected several main resistance genes and virulence genes of MRSA to analyze the internal mechanism of DMC’s inhibitory effect on MRSA. By analyzing the experimental results, we found that DMC showed a significant antibacterial effect and had a synergistic antibacterial effect with β-lactam antibiotic GEN. We explored the mechanism of DMC, and speculated that DMC compromised the vitality of MRSA by inhibiting the expression of PBP2a and related genes. In addition, the inhibitory effects of DMC on the expression of SEA and its related genes provided a distinct antibacterial mechanism. Our results suggest the potential of DMC to accelerate the healing of MRSA infections as a novel phytochemical compound. In the field of alternative medicine, DMC offers promises for the treatment of MRSA diseases. We plan to conduct further research on DMC to explore more physiological activities and antibacterial mechanisms of DMC against MRSA in the future.

## Figures and Tables

**Figure 1 ijms-22-03778-f001:**
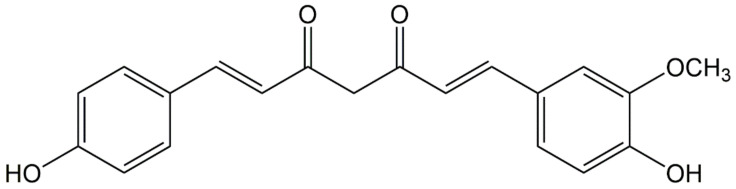
The chemical structure of demethoxycurcumin (DMC).

**Figure 2 ijms-22-03778-f002:**
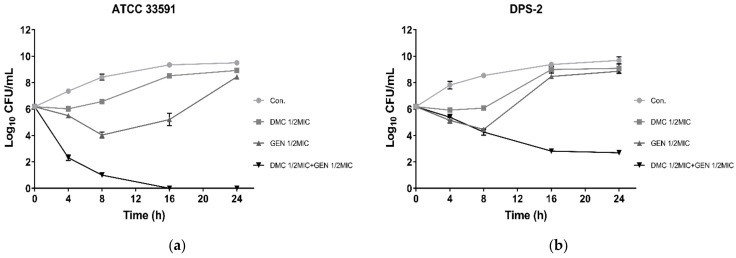
Time–kill curves of DMC and gentamicin (GEN) alone and in combinations against MRSA ((**a**) MRSA ATCC 33591; (**b**) MRSA DPS-2).

**Figure 3 ijms-22-03778-f003:**
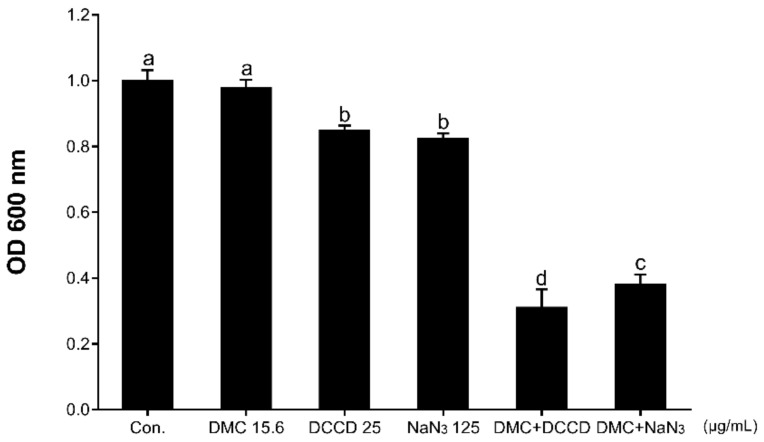
The effect of the ATP synthase inhibitors, DCCD (25 μg/mL) and NaN_3_ (125 μg/mL), on the susceptibility of MRSA strain ATCC 33591 to DMC treatment (15.6 μg/mL). The data are means ± SD of triplicate determinations. Different letters in each bars indicate significant statistical differences between treatments (*p* < 0.05).

**Figure 4 ijms-22-03778-f004:**
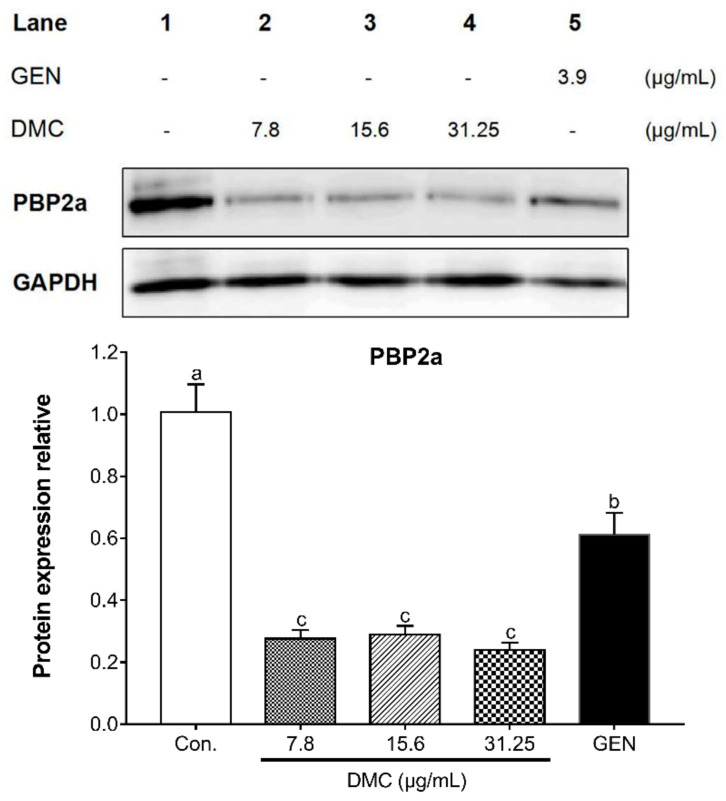
The effect of DMC and GEN on the expression of penicillin-binding protein 2a (PBP2a) in MRSA strain ATCC 33591, as analyzed by western blot. Loading differences were normalized by anti-GAPDH antibody. Lane 1, control (untreated); lane 2, DMC 7.8 μg/mL (1/8MIC); lane 3, DMC 12.5 μg/mL (1/4 MIC); lane 4, DMC 25 μg/mL (1/2 MIC); lane 5, GEN 3.9 μg/mL. The data are means ± SD of triplicate determinations. Different letters in each bars indicate significant statistical differences between treatments (*p* < 0.05).

**Figure 5 ijms-22-03778-f005:**
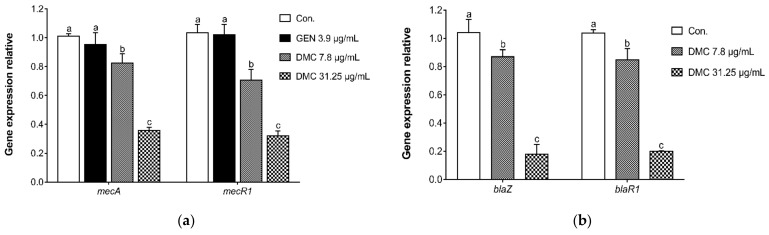
The effect of DMC and GEN on the mRNA expression of *mec* operon (**a**) and *bla* operon (**b**), as analyzed by qRT-PCR. ATCC 33591 were treated with serial dilution of DMC (7.8 μg/mL and 31.25 μg/mL) and GEN (3.9 μg/mL). The data are means ± SD of triplicate determinations. Different letters in each bars indicate significant statistical differences between treatments (*p* < 0.05).

**Figure 6 ijms-22-03778-f006:**
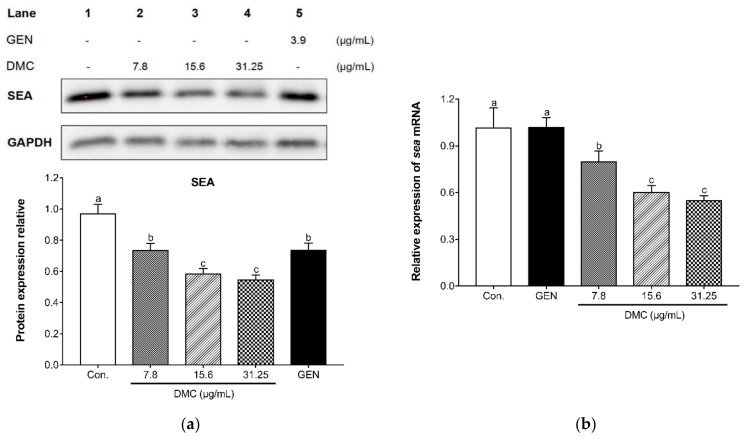
The effect of DMC on the protein (**a**) and mRNA (**b**) expression of staphylococcal enterotoxin A (SEA), as analyzed by western blot and qRT-PCR. ATCC 33591 were treated with serial dilution of DMC (7.8 μg/mL, 15.6 μg/mL and 31.25 μg/mL) and GEN (3.9 μg/mL). The data are means ± SD of triplicate determinations. Different letters in each bars indicate significant statistical differences between treatments (*p* < 0.05).

**Table 1 ijms-22-03778-t001:** Minimum inhibitory concentration (MIC) of oxacillin (OXA), ampicillin (AMP), gentamicin (GEN) and demethoxycurcumin (DMC) against eight *S. aureus* strains.

Strains	MIC (μg/mL)			
	OXA	AMP	GEN	DMC
ATCC 25923	0.9	0.9	0.9	62.5
ATCC 33591	125	125	7.8	62.5
DPS-1	62.5	62.5	125	62.5
DPS-2	0.9	1.9	125	62.5
CCARM 3090	250	31.25	125	62.5
CCARM 3091	1000	62.5	2000	62.5
CCARM 3095	500	31.25	250	62.5
CCARM 3102	250	62.5	500	62.5

Values are means of three independent experiments. ATCC, staphylococcal strains from the American Type Culture Collection; CCARM, staphylococcal strains from the Culture Collection of Antimicrobial Resistant Microbes; DPS, staphylococcal strains from the Department of Plastic Surgery, Wonkwang University Hospital.

**Table 2 ijms-22-03778-t002:** Minimum inhibitory concentration (MIC) and fractional inhibitory concentration index (FICI) of demethoxycurcumin (DMC), and gentamicin (GEN) against seven methicillin-resistant *Staphylococcus aureus* (MRSA) strains.

Strains	Agents	MIC (μg/mL)		FICI	Outcome
		Alone	Combination		
ATCC33591	DMC	62.5	3.9	0.306	synergy
	GEN	7.8	1.9		
DPS-1	DMC	62.5	7.8	0.375	synergy
	GEN	125	31.25		
DPS-2	DMC	62.5	7.8	0.250	synergy
	GEN	125	15.6		
CCARM 3090	DMC	62.5	7.8	0.375	synergy
	GEN	125	31.25		
CCARM 3091	DMC	62.5	7.8	0.375	synergy
	GEN	2000	500		
CCARM 3095	DMC	62.5	15.6	0.375	synergy
	GEN	250	31.25		
CCARM 3102	DMC	62.5	3.9	0.312	synergy
	GEN	500	125		

Values are means of three independent experiments. Index interpretation: <0.5, synergy; 0.5–0.75, partial synergy; 0.75–1, additive effect; 1–4, no effect; >4, antagonism.

**Table 3 ijms-22-03778-t003:** Sequences of oligonucleotide primers designed for qRT-PCR.

Gene	Primer Sequence
*mecA*	
Forward	5′-GCAATCGCTAAAGAACTAAG-3′
Reverse	5′-AATGGGACCAACATAACCTA-3′
*mecR1*	
Forward	5′-ACACGACTTCTTCGGTTAG-3′
Reverse	5′-GTACAATTTGGGATTTCACT-3′
*blaZ*	
Forward	5′-AGAGATTTGCCTATGCTTCA-3′
Reverse	5′-AGTATCTCCGCTTTTATTATTT-3′
*blaR1*	
Forward	5′-ACAATGAAGTAGAAGCCGATAGAT-3′
Reverse	5′-GTCGGTCAAGTCCAAACA-3′
*sea*	
Forward	5′-ATGGTGCTTATTATGGTTATC-3′
Reverse	5′-CGTTTCCAAAGGTACTGTATT-3′
*16S*	
Forward	5′-ACTCCTACGGGAGGCAGCAG-3′
Reverse	5′-ATTACCGCGGCTGCTGG-3′

## Data Availability

Personal information is included, so it was conducted for research only.

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
