# Peer review of "Study on Demethoxycurcumin as a Promising Approach to Reverse Methicillin-Resistance of Staphylococcus aureus"

_ijms, 2021, doi:10.3390/ijms22073778_

Round 1
Reviewer 1 Report
The manuscript as entitled as Study on Demethoxycurcumin as a promising approach to reverse methicillin-resistance of Staphylococcus aureus is well written. The article explains the comprehensive study of DMC's effect on MRSA. My only reservation is the lack of novelty and potent activity of DMC. Some of the typos and uniformity issues need to be addressed before publication.

Reviewer 2 Report
General: please make sure to define every abbreviation before use
Please make sure to use italics when discussing bacterial names
Abstract:
„thorny” please use a different word
phytochemical components
…for eliminating MRSA.
L27: that may cause
L29: Please consider including a following reference and discuss in more detail the clinical relevance of S. aureus/MRSA:
https://pubmed.ncbi.nlm.nih.gov/31052511/
L40: classical risk factors
L48: Because the cell wall…Gram-positive
L55: please also discuss that the most commonly used antibiotics taken for non-prescription use/self-medication use are beta-lactam agents and this also facilitates the emergence of methicillin-resistance, see reference:
https://pubmed.ncbi.nlm.nih.gov/32932630/
L61-65: please also note that another genetic determinant (mecC) is also possible to induce methicillin-resistance
L77: Zingiberaceae in italics
L106: and the curcuminoid
of the MIC values
L121: either re-designate the main section as „results and discussion” or remove the reference as it is not customary to have references in the Results
Please provide a color Figure on the preparation of the FICI assay.
wouldn’t it be more appropriate to calculuate some kind of other value (e.g., combination index (CI), see literature!
L174-176: do not forget PBP2c!!
L226-228: Please consider including the following reference:
https://www.nature.com/articles/s41598-020-74834-y
The discussion in its current form contains a lot of information that is already well-known. Instead, please try to better contextualize the significance of your findings in line with other recent experimental results. Try to restructure the discussion section accordingly.
